# Risks posed by invasive species to the provision of ecosystem services in Europe

Belinda Gallardo [1,2] ✉, Sven Bacher [3], Ana Marcia Barbosa[4], Laure Gallien [5], Pablo González-Moreno [6], Víctor Martínez-Bolea[1], Cascade Sorte[7], Giovanni Vimercati [3] & Montserrat Vilà [8,9]

Invasive species significantly impact biodiversity and ecosystem services, yet understanding these effects at large spatial scales remains a challenge. Our study addresses this gap by assessing the current and potential future risks posed by 94 invasive species to seven key ecosystem services in Europe. We demonstrate widespread potential impacts, particularly on outdoor recreation, habitat maintenance, crop provisioning, and soil and nitrogen retention. Exposure to invasive species was higher in areas with lower provision of ecosystem services, particularly for regulating and cultural services. Exposure was also high in areas where ecosystem contributions to crop provision and nitrogen retention were at their highest. Notably, regions vital for ecosystem services currently have low invasion suitability, but face an average 77% increase in potential invasion area. Here we show that, while high-value ecosystem service areas at the highest risk represent a small fraction of Europe (0-13%), they are disproportionally important for service conservation. Our study underscores the importance of monitoring and protecting these hotspots to align management strategies with international biodiversity targets, considering both invasion vulnerability and ecosystem service sustainability.

Since 1970, the capacity of nature to sustain human quality of life in the form of biodiversity, ecosystem functions, and services has been deteriorating[1–3]. The worldwide spread and establishment of invasive species is considered a major contributing factor to this decline[4–7]. Although the effects of biological invasions on native species and biodiversity have been extensively documented[7], evidence regarding their impacts on ecosystem services remains scattered in studies that focus on specific species or limited geographical areas[4,6,8], making it challenging to generalize findings to larger scales. The absence of spatial assessments of the risks posed by invasive species to ecosystem services can be partly attributed to the lack of harmonized accounting and mapping for multiple ecosystem services at large spatial scales[9].

This information gap is significant because the provisioning of ecosystem services varies across ecosystems, and the risks associated with biological invasions may differ depending on the type of service, whether regulating, provisioning, or cultural[10]. Obtaining a better understanding of the specific services that face the highest threats, and determining where we should concentrate conservation efforts, will aid in designing policies and regulations that address jointly biodiversity and ecosystem services. These two components are essential for nature's contributions to people and ensuring a good quality of life[1].

In this study, we leverage the increasing availability of invasive species risk assessments[11], species occurrence data, and recent

[1]Instituto Pirenaico de Ecología (IPE), CSIC, Avda. Montañana 1005, 50192 Zaragoza, Spain. [2]Biosecurity Initiative at St. Catherine's (BioRISC), Cambridge, UK. [3]Department of Biology, Unit Ecology & Evolution, University of Fribourg, Chemin du Musée 15, 1700 Fribourg, Switzerland. [4]Centro de Investigação em Ciências Geo-Espaciais (CICGE), Faculdade de Ciências da Universidade do Porto, Porto, Portugal. [5]Université Grenoble Alpes, Université Savoie Mont Blanc, CNRS, LECA, Grenoble, France. [6]Department of Forest Engineering, University of Cordoba, Campus de Rabanales, Crta. IV, km. 396, 14071 Córdoba, Spain. [7]Department of Ecology and Evolutionary Biology, University of California, Irvine, CA, USA. [8]Estación Biológica de Doñana (EBD), CSIC, Avda. Américo Vespucio 26, 41092 Sevilla, Spain. [9]Department of Plant Biology and Ecology, University of Sevilla, 41012 Sevilla, Spain. ✉e-mail: belinda@ipe.csic.es

advances in ecosystem services mapping[9], to provide a comprehensive spatial evaluation of the risks posed by biological invasions to the provisioning of ecosystem services at a continental scale. Our specific aims are to quantify the current observed risks posed by invasive species to the provisioning of ecosystem services, and map spatial patterns of future risks. This study is based on the premise that risks varies across regulating, provisioning, and cultural services, which are typically delivered by different ecosystems exhibiting contrasting levels of invasion[6,7,9].

The severity of the impacts associated with natural risks strongly depends on the level of vulnerability and exposure to the hazard[12]. In this study, exposure is measured based on the current and/or future potential presence of invasive species, assuming a scenario where they can occupy the most accessible and favorable parts of their ecological niche. Vulnerability is determined by the level of provisioning of ecosystem services that are known to be susceptible to each invader, with both very high and very low provisioning areas considered vulnerable to biological invasions[13]. For instance, human-altered habitats are highly exposed to invasions because of their high levels of disturbance and propagule pressure[14,15]. This is frequently the case for invasive plants, whereas invasive animals tend to be more evenly distributed in natural and human-altered environments[15,16]. However, human-altered habitats are typically unable to deliver a wide variety of ecosystem services[1], therefore displaying limited vulnerability. In contrast, well-conserved areas largely contributing to many different services may be less exposed to the current and potential threats of biological invasions, but are highly vulnerable, making any future changes significantly impactful[17].

This study focuses on Europe (except Russia) due to data availability and the significance of advanced environmental policy regulations ranging from national to pan-continental scale[18]. This ensures that our results can inform policy decisions, especially regarding the implementation of the European Regulation on Invasive Alien Species, which aims to prevent, minimize, and mitigate the adverse effects of biological invasions on biodiversity and related ecosystem services (EU Regulation no. 1143/2014). Specifically, we aim to provide an spatial evaluation of the risks posed by 94 invasive species to seven ecosystem services, encompassing habitat maintenance, nitrogen retention, soil retention and flood control, crop and timber provision, as well as the provision of outdoor recreation. Overall, this study provides important insights into the risks posed by biological invasions to ecosystem services generally, and can inform policy decisions regarding invasive species management and biodiversity conservation efforts.

## Results and discussion
### The current exposure of ecosystem services to invasion
We focused our research on 32 terrestrial plants, 29 terrestrial animals, 20 freshwater animals, and 13 freshwater plants, all of which are included in the European List of Invasive Species of Union Concern (the Union List) (Supplementary Data 1). This list comprises non-native organisms that, based on available scientific evidence, are known to have a significant adverse impact on biodiversity or related ecosystem services at a pan-continental scale, and thus constitute a priority for management. Currently, the Union List contains 81 continental invaders (as of January 2023). Additionally, we included 13 species that are under consideration for listing by member states. This selection ensures the ongoing relevance of our study, as species under consideration may be added to the Union List in the foreseeable future.

The impacts of invasive species of European concern on biodiversity, ecosystems, and associated services have been fully risk-assessed in a systematic manner that allows comparison across species[11]. We utilized these risk assessments to identify the specific invaders potentially affecting each of the seven services evaluated. While invasive species may themselves provide services such as food, timber, protection against soil erosion and water purification, our

analysis focuses solely on negative impacts. This is because our study focuses on a specific set of species prioritized for management, where their harmful effects are considered to outweigh any potential benefits. Moreover, the benefits of invasive species have not yet been systematically reviewed in risk assessments[19], which prevents a full objective assessment of positive and negative outcomes. In the future, the systematic description and evaluation of benefits provided by invaders to ecosystem services provision can aid in identifying conflicts of interests and promoting dialog among stakeholders[19].

Across the 658 potential combinations between the 94 invasive species and seven ecosystem services investigated, we confirmed a total of 269 potential impacts, accounting for 41% of the combinations (Supplementary Data 1). Naturally, not all invaders possess the capacity to impact every ecosystem service. For instance, an aquatic animal is unlikely to affect crop or timber production, whereas a terrestrial plant has limited repercussions for water quality. Consistent with global assessments[7], the most frequently affected services were the provision of daily outdoor recreation (70 species), habitat maintenance (57 species), crop provision (41 species), soil and nitrogen retention (43 and 31 species, respectively). Conversely, the ecosystem services with the fewest number of impacts included timber provision (14 species) and flood control (13 species).

We quantified current exposure of ecosystem services to invasion using the total number of species known to impact each service present within each 10×10 km grid cell (see Supplementary Data 2 for the sources of and number of species occurrences used). We utilized the ecosystem services accounts provided by the European Environmental Agency[9] for seven regulating, provisioning, and cultural ecosystem services. Regulating services included: (i) Habitat maintenance to support species populations[20], nitrogen retention by natural ecosystems[20], soil retention[20] and flood control[21]. Provisioning services encompassed the contribution of ecosystems to crop production[21] and timber provision[21]. The representation of cultural services focused on outdoor daily recreation[22] (Supplementary Table 1). We used maps reflecting the "potential" for ecosystem service provision, which is a measure based on the basic biophysical characteristics of ecosystems. This is less susceptible to change than the "realized" provision of ecosystem services that incorporates the demand for the service. The potential provision of ecosystem services has remained relatively stable in Europe within the last decade, whereas the demand has steadily increased resulting in a growing need by society not fully covered by natural ecosystems[9,20–22], a situation that may be worsened by the spread of invasive species. To investigate vulnerability, we reclassified maps of ecosystem service provision into three categories: low (lowest 20% of service values), medium (central 20–80% of service values), and high (upper 20% of service values; Supplementary Fig. 4).

We observed that the current exposure to biological invasions is more pronounced in areas with low provisioning of ecosystem services, but only for regulating and cultural services such as habitat maintenance, flood control, and outdoor recreation (Fig. 1A–D). Interestingly, we found that the current exposure to invasive species was significantly higher in areas where ecosystem contributions to crop provision and nitrogen retention were at their highest (Fig. 1E, F). Pairwise differences in exposure across low, medium, and high provisioning areas were highly significant in all cases (Supplementary Table 2). These patterns remained consistent when using alternative thresholds to classify ecosystem services into low, medium, and high provisioning areas, with only minor deviations in the mean number of invasive species per grid cell (5.8–6.6%, Supplementary Fig. 5). General patterns observed for current exposure were maintained for future potential exposure (Supplementary Fig. 6).

Geographically, the current exposure of Europe to the 94 invasive species of concern is primarily concentrated in Western Europe, specifically in countries such as the UK, the Netherlands, France, Belgium

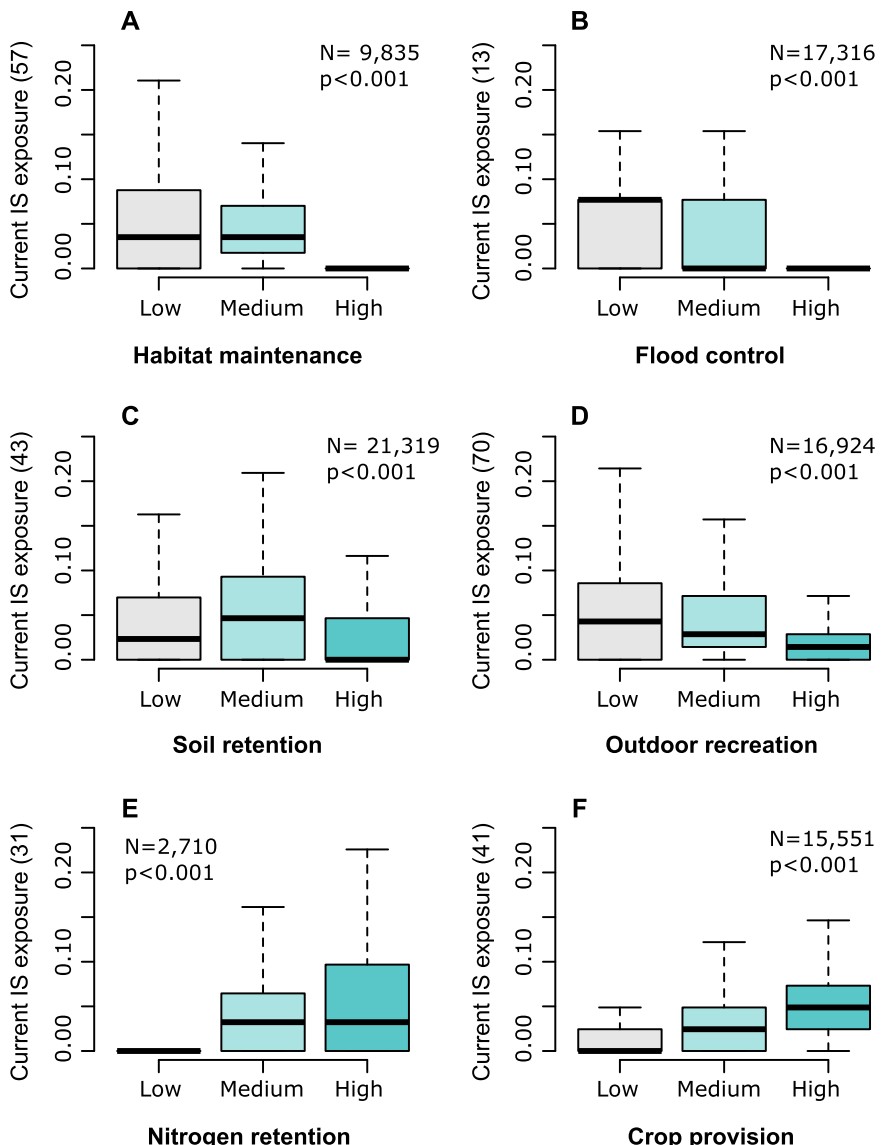

**Fig. 1 | Current exposure of ecosystem services to the invasive species most likely to affect them. A** Habitat maintenance. **B** Flood control. **C** Soil retention. **D** Outdoor recreation. **E** Nitrogen retention. **F** Crop provision. Exposure is measured as the number of invasive species (IS) per 10 × 10 km grid cell currently present in areas delivering Low, Medium and High levels of ecosystem services. To account for variations in the number of invasive species affecting each service (indicated in brackets in each panel), exposure values were rescaled to a range of 0-1. Box-plots depict the median, interquartile interval, and 1.5× interquartile range. N in the upper right corner correspond to the number of grid cells in Europe with valid ES and invasive species information. *P*-values correspond to Welch one-way ANOVA (two-sided). Pairwise Tukey HSD post-hoc comparisons can be consulted in Supplementary Table 2.

and Ireland (Fig. 2A). This concentration aligns with findings commonly reported in the literature and can be attributed to a combination of mild climatic conditions, intense human influence and a long history of overseas trade and travel[13,14,17,23].

**The future exposure of ecosystem services to invasion**
We employed species distribution models to compute favorability maps, i.e. maps showing the areas most favorable for the establishment of invasive species under current climate conditions. Models used global occurrence data of species and a range of predictors including climate, elevation, and human accessibility, which accounts for the transportation and establishment of invaders facilitated by human activities, i.e. propagule pressure[14,17] (see Supplementary Data 3 for the results of model cross-validation). We then classified the resulting favorability maps into three categories: 1 (favorability ≲20%), 2 (20–80% favorability), and 3 (favorability ≳80%). By assuming that

invasive species can spread across all sites with highly favorable conditions, these maps allowed us to predict the future potential exposure of ecosystem services to biological invasions.

Our findings reveal a substantial increase in the future exposure of ecosystem services to invasive species of concern in Europe, particularly along coastal areas, and the Atlantic and Continental biogeographic regions (Fig. 2B). The extent of range expansion varied among the species analyzed (Fig. 3), but, on average, the suitable area for invasion increased by 77% compared to the currently occupied area (Fig. 2C, differences between Current and Potential are significant according to a paired *t*-test: $t = -7.14$, df = 93, $P < 0.001$). Moreover, when employing a less conservative threshold, the potential for expansion increased significantly. For example, using a favorability threshold of ≳70%, which is still remarkably high, resulted in a potential range expansion of 163%, on average, beyond the current area occupied by our focus invasive species (Supplementary Fig. 9).

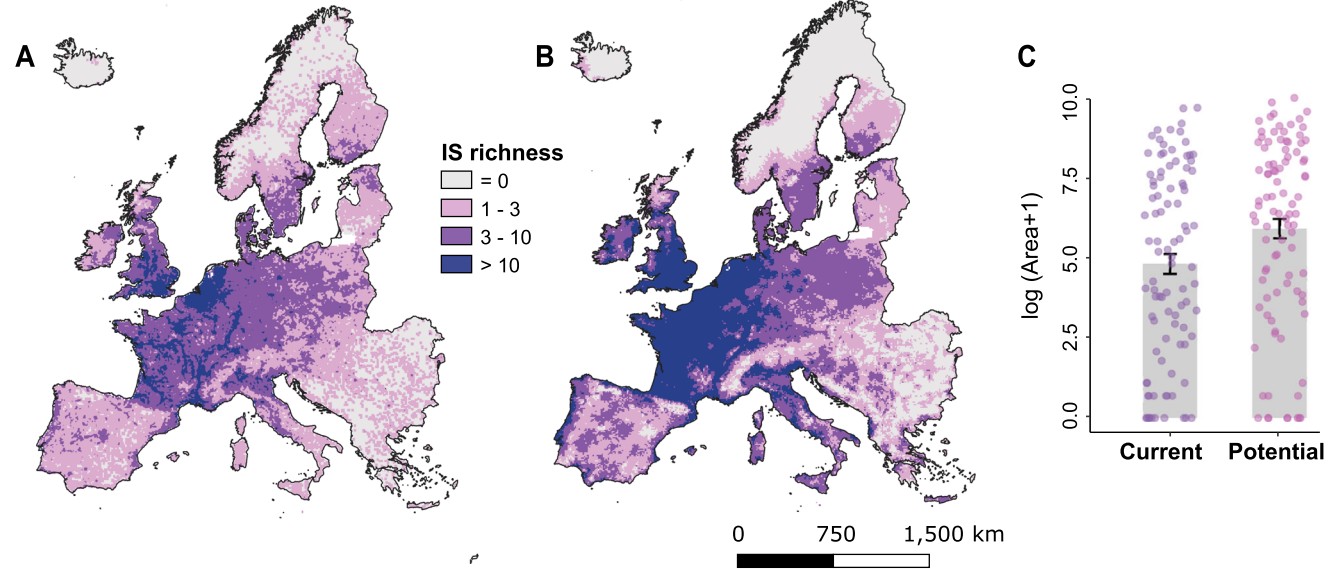

**Fig. 2 | Current and potential exposure to 94 invasive species regulated in Europe. A** Current exposure is measured based on the real number of invasive species currently present using occurrences. **B** Potential exposure is based on species distribution model predictions of invasive species establishment. **C** Barplot showing the average area (+SD) under current and potential exposure to invasive species of concern in Europe, respectively. Please note that, to facilitate visualization, the area has been log-transformed. Differences between Current and Potential are highly significant according to a paired Student's $t$ test ($t = -7.14$, df = 93, $P < 0.001$).

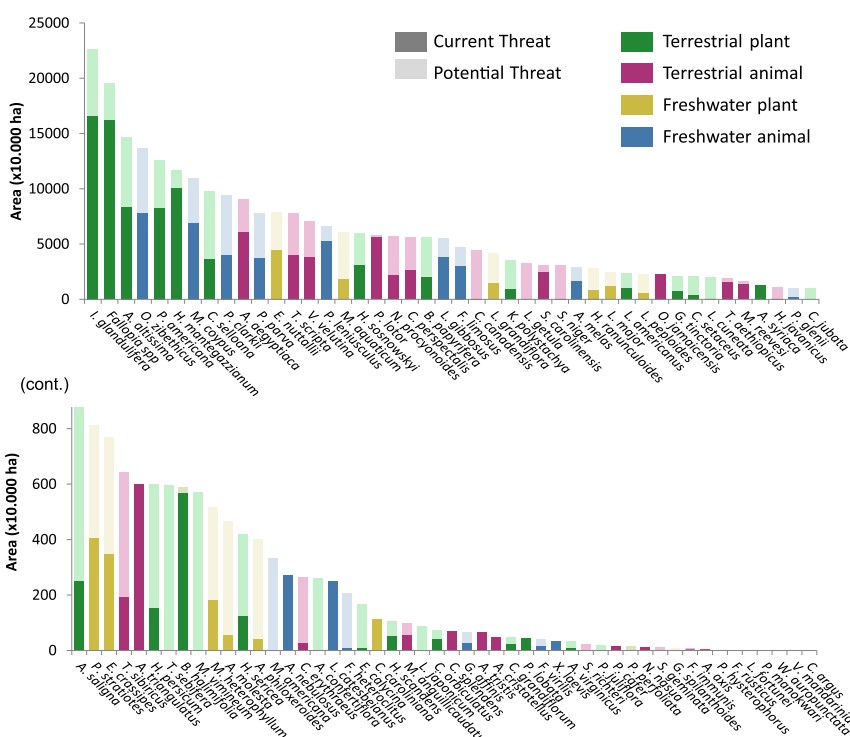

**Fig. 3 | Current and potential distribution (=range expansion) of 94 invasive species of concern in Europe.** Colors depict four major groups of organisms: terrestrial plants (green), terrestrial animals (pink), freshwater plants (yellow) and freshwater animals (blue). In dark shade, the total area currently occupied by each species. In light shade, the additional area that is predicted as favorable for the establishment of the species under current climate and accessibility conditions. Species are ordered decreasingly by the total invasible area (current + potential). The lower panel is the continuation of the upper panel.

## Risks posed by invasive species to ecosystem services

To assess the risks posed by invasive species to ecosystem services, we employed a two-fold approach, considering both the favorability of species occurrence (exposure) and the vulnerability of ecosystem services[12,13]. First, we assumed that invasive species have a greater impact in areas with high favorability, potentially leading to increased abundance and *per capita* impact[24]. Second, within areas highly favorable to a specific invader, we considered that impacts depend on the provision of the ecosystem services sensitive to that particular invader. We thus defined a total of nine risk categories by combining

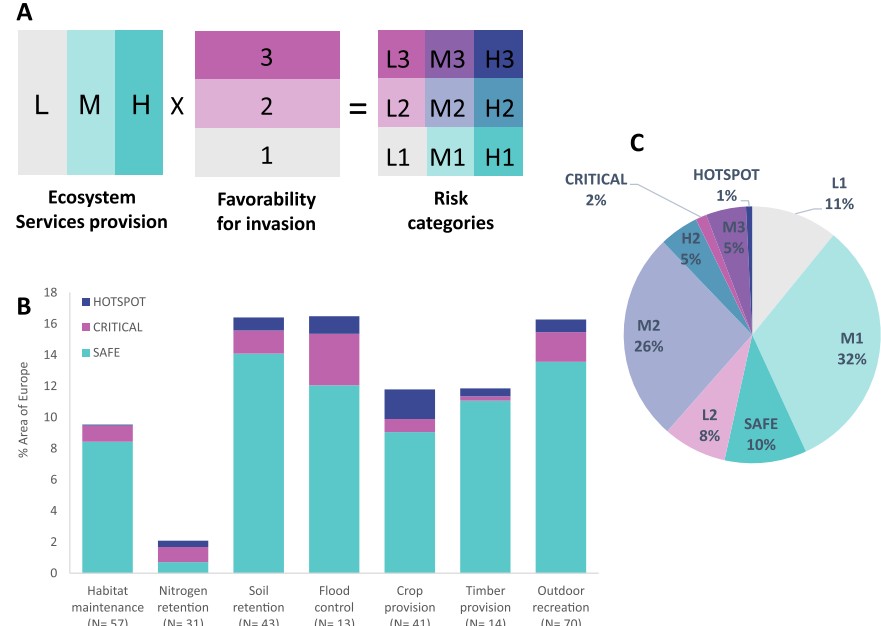

**Fig. 4 | Potential risk posed by invasive species to the provision of seven ecosystem services in Europe. A** Nine risk categories resulted from the combination of three levels of ecosystem services supply (L, M, and H) and three levels of invasive species favorability (1, 2, and 3). **B** Percentage of the area in Europe classified into the three main risk categories (safe, critical and hotspot) for each of the seven services investigated. Data averaged over the total number of invasive species affecting each service, which are indicated in brackets. **C** Percentage area classified in each of the 9 risk categories across the 94 invasive species and seven ecosystem services investigated.

the provision of ecosystem services (Low, Medium, and High categories) and the favorability for invasion (1, 2, and 3 categories; Fig. 4A and Supplementary Data 4). This sets our research apart from other studies that have relied on general patterns of invasive species richness. By integrating both exposure and vulnerability, we aimed to provide a comprehensive understanding of the risks posed by invasive species to essential ecosystem services. In our analysis, we particularly focused on three corners of the risk matrix, which we interpreted as Safe (H1), Critical (L3), and Hotspots (H3), further described below (Fig. 4B and Supplementary Data 4).

Across all possible combinations of invasive species and ecosystem services, an average of 10.2 ± 5.8% of the European area was classified as Safe (Fig. 4C). This percentage varied across individual species and ecosystem services, ranging from 0% to 20%. Safe areas, characterized by high service supply and low potential exposure to invasion, represented the dominant category for six of the seven ecosystem services analyzed. The exception was nitrogen retention by natural ecosystems (a proxy for water purification), which exhibited high retention combined with low potential exposure to invasion in only 0.4% of Europe (Fig. 4B).

In comparison, Critical areas, characterized by low service provisioning and high potential exposure to invasion, accounted for an average 1.4 ± 2.0% of the European area (range 0–9%, Fig. 4C). Low ecosystem services provisioning has been shown to decrease the social-ecological resilience of natural ecosystems to disturbance[25]. As a result, Critical areas can be particularly vulnerable to the adverse consequences of invasions. For instance, in areas where natural ecosystems have limited soil retention capacity, the colonization of invasive plants that contribute to erosion can significantly disrupt the stability of the ecosystem[13].

Hotspots, defined as areas with both high potential exposure to invasion and high service supply, accounted for an even smaller portion: only 0.8 ± 1.8% (Fig. 4C). The extension of hotspots ranged between 0 and 13% of the European area, and were particularly important for crop provision, flood control, and soil retention (Fig. 4B).

The neighboring risk categories (M3 + M2 + H2) encompass an average of 36.4 ± 16.6% of the European territory (ranging from 1% to 90% depending on the species and services considered, Supplementary Data 4). However, this situation may change in the future as invaders continue to expand their range and new invasive species enter the continent, which are not yet included in the Union List. While climate change was not explicitly considered in this study, it will pose unprecedented challenges to the delivery of ecosystem services[26]. Previous studies incorporating climate change into predictions have confirmed the increasing threat posed by invasive species to the provision of ecosystem services, particularly in high altitude and latitude areas[13,17].

Sensitivity analysis using alternative thresholds for service provisioning showed variation in the classification of Hotspots (0.6–1.8% of Europe) and Critical areas (0.7–2.3% of Europe), while Safe areas ranged from 6.4% to 17.7% of Europe (Supplementary Fig. 11).

## Spatial distribution of the risks posed by invasive species

The spatial distribution of impact hotspots and critical areas (Supplementary Fig. 12) exhibited notable variations across different ecosystem services. Geographically, invasive species of European concern pose a threat to habitat maintenance predominantly in high latitude and altitude areas within the Boreal, Alpine, and Atlantic bioregions (Fig. 5A). Habitat maintenance was affected by 57 of the 94 invasive species investigated (60%), most of them terrestrial plants that alter habitat structure and suppress native species, rapidly dominating the landscape and affecting community composition[27,28]. It has traditionally been assumed that cold environments are less susceptible to invasion due to their extreme climatic conditions and limited accessibility[29]. However, research has demonstrated an increasing occurrence of invasive plants, insects, and pathogens in cold environments, facilitated by human activities such as roads and land-use changes, a trend expected to further accelerate with ongoing climate change[30].

The Atlantic bioregion in Europe emerges as a hotspot for flood control (Fig. 5B). Forested areas in Europe exhibit high potential for

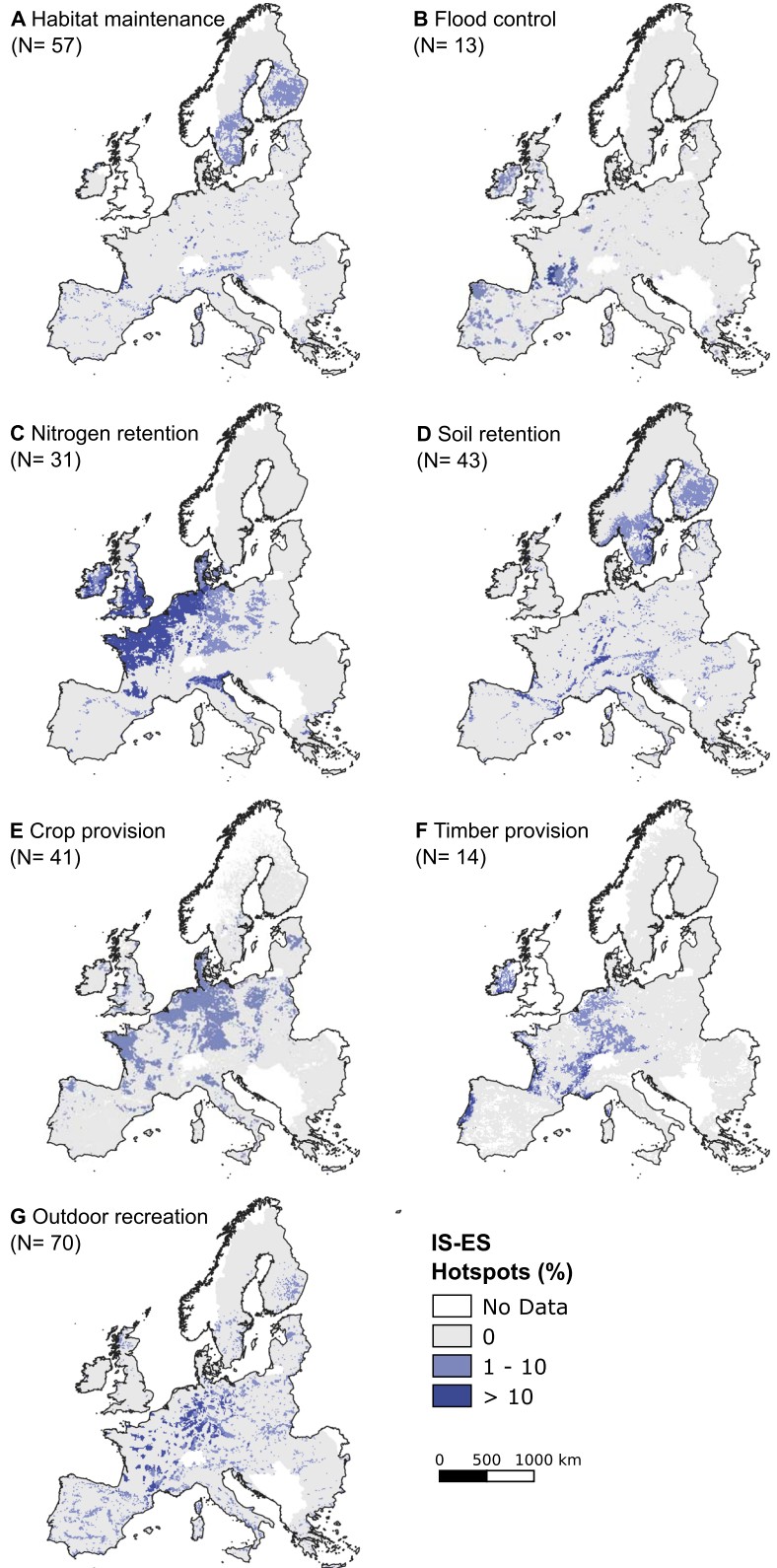

**Fig. 5 | Spatial distribution of hotspots for invasive species risks to the provision of ecosystem services. A** Habitat maintenance. **B** Flood control. **C** Nitrogen retention. **D** Soil retention. **E** Crop provision. **F** Timber provision. **G** Outdoor recreation. Values represent the % of invasive species known to negatively affect each specific ecosystem service (indicated in brackets in each panel), that are predicted to be present in the near future within high-provisioning areas for ecosystem services. Hotspots in dark blue are areas that exhibit both high delivery of ecosystem services and high potential exposure to the invasive species that may affect them.

flood control, while the main agricultural plains with extensive human development display lower values[21]. A total of 13 invasive species were reported to aggravate the effects of floods, primarily consisting of aquatic plants that can impede sediment flow, obstruct canals, and reduce flood attenuation[4]. Additionally, certain invasive animals can contribute to channel collapse, altering the landscape's geomorphology and hydraulics[31]. The importance of such impacts to hydrology is expected to escalate due to growing demands for water resources, exacerbated by human population growth and global environmental change[31].

The risk assessments reviewed for this study evidenced the potential of 31 invasive species to alter nutrient cycles, consequently affecting water quality, particularly in regions with high nitrogen retention demands, such as central Europe (Fig. 5C). Invasive aquatic plants reduce oxygen levels, accelerate decomposition rates, disrupt nutrient cycling, and contribute to eutrophication, especially during massive die-offs[6,32]. Invasive animals can also contribute to eutrophication through depositions, increased turbulence and turbidity, which results in the resuspension of nutrients[6,32]. Thus invasive species reduces the ability of ecosystems to retain and process excessive nutrients derived from human activities, which is not only an economically valuable service, but also indirectly supports other services such as recreation and human health[33].

Hotspots for soil retention are predominantly found in the Boreal, Alpine, and Continental bioregions (Fig. 5D). Among the 43 invasive species affecting this ecosystem service, terrestrial plant invaders (26 species) play a significant role in reducing soil retention. Invasive plants can change soil moisture levels, restrict the growth of grass due to canopy dominance, leave the soil bare during winter, and alter soil properties such as porosity, drainage, litter accumulation, and nutrient cycling[34]. Additionally, certain invasive plants can increase the frequency and intensity of fires, leading to further changes in soil formation and maintenance[35]. Invasive animals can also increase erosion by eating and uprooting plants, or by disturbing and burrowing soils[4,6].

Our findings indicate that the effects of invasive species on crop provisioning are likely to be most pronounced in the Atlantic and Continental bioregions (Fig. 5E). These regions contribute significantly to crop yields due to lower reliance on irrigation and mineral fertilizers (Supplementary Fig. 4E). In particular, 57 of the 94 species investigated have reported impacts on crop provision through the direct consumption of plants, grain, and fruits, competition for light, water, and nutrients, the production of allelopathic substances, and direct destruction[6,36]. While Europe has benefited from the widespread use of plant protection measures (e.g., pesticides, herbicides) and cultivation techniques like crop rotation minimize the incidence of crop pests, it is important not to underestimate the impacts of invasive species, which can be more significant than those of climate change for many crops[26]. Additionally, climate change may reduce the effectiveness of plant protection measures[37], further exacerbating the impacts of biological invasions on crop provisioning.

The Atlantic, Continental, and parts of the Mediterranean bioregions, which are known for their high ecosystem contribution to timber provisioning, also coincide with the concentration of impact hotspots (Fig. 5F and Supplementary Fig. 4F). Among invasive species of European concern, 14 have been identified as affecting timber production. These impacts primarily stem from terrestrial plants that compete directly with timber-producing species, increase their vulnerability to pathogens and pests, or make access to plantations more difficult[38]. Invertebrate pests can also have detrimental effects on tree growth, leading to the death of mature trees or impacting nursery operations. Additionally, vertebrates consume forest products such as nuts, fruits, and seeds, further impacting timber provisioning[39].

Outdoor recreation opportunities provided by nature can be significantly affected by biological invasions[6,22]. As a result, the impact hotspots on outdoor recreation are well distributed across Europe,

with a higher concentration in the Continental bioregion (Fig. 5G). Among the 94 invasive species investigated, a majority of 70 species have been found to diminish the recreational quality of the landscape. These impacts include trampling, fouling, aggressive behavior, causing dermatitis, lacerations, and allergies, as well as restricting access to and normal development of outdoor activities[4,40].

## General discussion

This study provides a comprehensive cross-taxon spatial analysis of the risks posed by invasive species to the provisioning of ecosystem services, including habitat maintenance, nitrogen and soil retention, ecosystem contribution to crop and timber provision, and daily outdoor recreation. Our findings reveal an important disparity between the spatial distribution of invasive species and the most vulnerable areas in terms of ecosystem service provisioning, particularly for regulating and cultural services. This observation aligns with previous studies highlighting the highest density of invasive species and their impacts in urban, cropland, and grassland ecosystems in Europe[23], as well as at the global scale[15]. Indeed, it is well-established that human-altered habitats are especially susceptible to invasions due to their elevated levels of disturbance and propagule pressure[14], but see[16]. Such habitats are also known to provide fewer ecosystem services, or more concentrated in provisioning services, such as energy, food, and timber[1].

While there is ample room for expansion of invasive species in Europe, our study demonstrates that most areas with high or moderate ecosystem service provisioning have low accessibility and climatic suitability for invasive species, thereby limiting exposure. This finding further emphasizes the spatial mismatch between the current distribution of invasive species and the areas delivering essential ecosystem services. Results complement earlier research that revealed lower-than-expected occurrence of invasive species within Europe's network of protected areas, primarily attributed to their low accessibility, and consequently low propagule pressure[17]. It is important to note that despite their lower exposure, the potential effects of invasive species on biodiversity and ecosystem conservation in vulnerable areas could still be substantial due to their high conservation value[5,17].

Our cross-taxonomic quantification and spatial assessment of the risks posed by invasive species to multiple ecosystem services represents a significant progress compared to previous studies that have primarily focused on describing impacts based on well-known examples[4–6,8,27]. However, it is important to acknowledge that our assessment has limitations in fully capturing the magnitude of impacts and their context dependence. The mechanisms driving the impacts of invasive species on ecosystem services are complex, and empirical evidence is often limited. The presence of an invasive species does not necessarily guarantee an impact on ecosystem services, as factors such as invader density, microhabitat, and management measures can influence the outcomes. Uncertainty in our research arises from several factors, including the association between invasive species and ecosystem services, the modeling of species, the production of ecosystem services, map resolution, and the thresholds used to define high exposure and vulnerability levels. We have incorporated sensitivity analyses to address and quantify part of this uncertainty, which have not indicated significant variability in our results. As the availability of quantitative information regarding the relationship between the abundance of invasive species and their impacts on ecosystem functioning and services continues to grow, future research should build upon our initial results. This research should explore how changes in one ecosystem service, due to the presence of invasive species, have cascading effects on other services across scales e.g. ref. 41. Additionally, it is important to consider the cumulative or synergistic consequences of multiple invasive species and how they collectively impact ecosystem services. Indeed, our analysis made no assumptions regarding the additive or synergistic impacts of co-

occurring invasive species. However, interactions deserve further investigation, as the combined presence of multiple invasive species could exacerbate their individual impacts, potentially leading to a threshold where ecosystem functioning becomes severely compromised. By considering these connections, we can gain a more comprehensive understanding of the broader socio-ecological consequences of biological invasions.

In conclusion, with limited resources available for management, it is crucial to prioritize monitoring and conservation efforts in areas that are likely to have the highest ecosystem services at risk[4,6]. Our findings indicate that these areas represent only a small proportion of Europe, a finding that should be tested across scales to determine generality. However, this does not mean that the control of invasive species should be ignored in sites with intermediate risk, which cover a much larger area according to our study. To assist in this prioritization process, we have provided risk tables and maps for all the current and prospective terrestrial and aquatic priorities in the Union List that can be used to ensure the protection of key ecosystem services, regardless of the underlying biodiversity. This spatially explicit information is fundamental to support the EU strategy for the future management of biological invasions recently proposed by Roura-Pascual et al.[42]. Ultimately, our findings can contribute to the 2030 Biodiversity Strategy that seeks to restore Europe's biodiversity for the benefit of people, climate and the planet. The modeling approach used in this study has broad applicability to other invasive species, regions, and socio-economic contexts worldwide, which would contribute to the development of quantitative future invasion scenarios[42,43]. This is important because Europe is a highly anthropogenic continent, and therefore, results from less disturbed areas of the world may differ. Spatial risk assessments are of strategic importance in guiding actions aimed at meeting the targets set by international agreements like the Kunming-Montreal Global Biodiversity Framework, focused on protecting biodiversity and the associated ecosystem services for the benefit of both people and nature. By targeting conservation efforts towards priority areas, we can maximize the effectiveness of management actions and ensure the long-term sustainability of ecosystem services.

## Methods

A conceptual summary of the workflow employed in this study to assess the risks posed by invasive species to the provision of ecosystem services can be found in Supplementary Fig. 1.

### Selection of invasive species of concern

The EU Regulation 1143/2014, on the prevention and management of the introduction and spread of invasive alien species (https://ec.europa.eu/), establishes a Union List that, based on available scientific evidence, are likely to have a significant adverse impact on biodiversity, ecosystem services, human health or the economy. The inclusion of species on this list is based on comprehensive risk assessments conducted by independent experts and reviewed by the EU Scientific Forum. In this study, we utilized the information from these risk assessments to identify the risks posed by invasive species to ecosystem services. Our analysis focused on the 94 non-marine organisms that have undergone full risk assessment and have been identified as management priorities at the European scale (last accessed August 2022, https://circabc.europa.eu/). This includes 33 terrestrial plants, 31 terrestrial animals, 18 freshwater animals, and 12 freshwater plants (Supplementary Data 1). We note that 13 of these species are still under consideration for inclusion in the Union List. However, we adopted an inclusive approach in our study in an attempt to maximize the future utility of our analyses.

We retrieved information on potential impacts to ecosystem services from questions about the impact of each species on biodiversity, ecosystem services and the economy. The magnitude of impacts was often quantified in a Minor to Massive scale (or Low to

High, depending on the type of protocol used, see Supplementary Data 1). Consequently, we adopted a precautionary approach and categorized potential impacts as either "yes" or "no". It is important to note that: (1) the magnitude of impacts can vary depending on the specific context, making it difficult to extrapolate values to the entire affected area, (2) assessors' information sometimes lacked the necessary details to estimate impact magnitudes per service, and (3) certain invaders are not yet present in Europe, leading to uncertainty regarding their potential impacts. Consequently, we considered the mere presence of a species as potentially impacting ecosystem services based on reported impacts elsewhere.

### Species presence and pseudo-absence data

We obtained worldwide occurrence data for the 94 invasive species from the Global Biodiversity Information Facility (GBIF, https://www.gbif.org; see Supplementary Data 2 for DOIs, CC-BY-NC). Relying on global occurrence data is essential, ensuring coverage of the full environmental niche of species and avoiding underestimation, especially for species not yet in Europe or recently established[44,45]. However, it must be noted that GBIF records do not differentiate casual from established self-sustaining populations, so that they may overestimate the current spread of invaders. However, it is preferable to overestimate rather than underestimate or bias the spread of invasive species, since casual populations may still exert the same impacts on biodiversity and ecosystem services.

GBIF data was imported into R and cleaned of inaccuracies with the 'scrubr' R package v 0.3.2. We further filtered out records whose coordinate uncertainty was higher than 50 km, to minimize spatial error while avoiding a large loss of records. We included additional occurrences from: (1) the European Alien Species Information Network (EASIN; https://easin.jrc.ec.europa.eu/easin, CC BY 4.0), (2) several original sources compiled in ref. 17, and (3) a recent data paper compiling invasive occurrence records from additional sources[46] (Supplementary Data 2).

Species occurrence records were gridded to a 10 arc-minute spatial resolution, which is approximately 11x11 km at the centroid of Europe, using the 'fuzzySim' R package v 3.8. This function thins (i.e. rarefies) occurrences to one maximum point per pixel, which reduces spatial autocorrelation induced by survey bias[47]. In total, for the 94 invasive species investigated, we gathered >9 million occurrence records that after cleaning and gridding resulted in 310,556 data points (see Supplementary Data 2 for a breakdown by species). Individual maps of occurrences in Europe can be consulted in Supplementary Fig 2. The number of analyzed species per grid cell in Europe is mapped by major taxonomic groups in Supplementary Fig. 3.

Pseudo-absences were generated by randomly selecting separate 10-arc-minute pixels, in the same number of each species' presences, outside the presence pixels but within a 200-km spatial buffer around them. This aims to restrict the choice of pseudo-absences to potentially reachable areas in regions where the species has been surveyed, thus conferring pseudo-absences the same survey bias as presences, which is key to improve the transferability of models[48,49]. This spatial selection was done using the 'terra' R package v 1.5.21.

### Spatial provision of ecosystem services

We utilized ecosystem service proxies from two platforms of the European Environmental Agency (EEA): the Mapping and Assessment of Ecosystems and their Services (MAES, https://biodiversity.europa.eu/ecosystems/maes), and the Accounting for ecosystems and their services in the European Union (INCA, https://ecosystem-accounts.jrc.ec.europa.eu/about-inca). These two platforms provide comprehensive reports and maps for nine ecosystem services, employing a consistent methodology based on European data and statistics[9]. The advantage of using EEA data and maps lies in their methodological and practical consistency, making them suitable for policy development

and future assessments. We selected seven ecosystem services for our analysis based on the number of invasive species with reported impacts: habitat maintenance, nitrogen retention, soil retention, flood control, crop production, timber provision, and outdoor recreation. Detailed descriptions, sources, and metadata of the maps can be found in Supplementary Table 1. All ecosystem services maps were obtained from the Joint Research Centre (https://data.jrc.ec.europa.eu/dataset, last accessed July 2022, CC BY 4.0). Please note that the reference year for the mapping of ecosystem services is 2012, which means that the level of provision incorporates the historical influence of various threats, including biological invasions.

We calculated average values for each ecosystem service proxy by aggregating data over the 10×10 km reference grid provided by the EEA, using R package 'exactextractr' v 0.6.1. The maps of all proxies were then rasterized using this grid as a template, ensuring that each pixel exactly matches one EEA grid cell. To facilitate analysis and interpretation, the values of each proxy were classified into three categories using the upper and lower 20% percentiles: low provisioning (≤20% of data), medium provisioning (20–80% of data), and high provisioning (≥80% of data) (Supplementary Fig. 4). The classification employed narrower ranges at the extremes to better differentiate areas with very low or very high provisioning levels.

Using categories of ecosystem service provision instead of raw values is justified for several reasons. First, the classification allows for a more intuitive interpretation of the data, as it simplifies complex and potentially variable values into distinct categories. Second, the absolute thresholds for high or low provisioning are not well-defined or universally agreed upon. By using percentiles, such as the upper and lower 20%, we can create categories that capture the relative differences in provisioning levels within the dataset. Third, the use of categories allows us to identify areas of particularly high or low provisioning relative to the overall distribution, providing valuable insights for management and decision-making.

## Current exposure of ecosystem services to invasion

We generated exposure maps by stacking the cumulative number of invasive species potentially impacting each ecosystem service within 10×10 km grid cells (Supplementary Data 2 and Figs. 2 and 3). Current exposure was quantified as the total number of invasive species affecting a specific ecosystem service within each grid cell, without considering their abundance. Several factors justify our approach. First, abundance may vary seasonally across their distribution range. Second, the relationship between abundance and impact is non-linear; in some cases, there can be thresholds in abundance before there are deleterious impacts, whereas in others even low abundances can disrupt ecosystem functions[50]. Third, information about abundance data is rarely available at large spatial scales like the one used in this study. Fourth, abundance of invasive species generally increases over time[51], which means that using current or historical abundance would underestimate the potential impacts of invaders. Considering these complexities, we believe that our approach is appropriate for an initial approximation.

We used Welch one-way ANOVA for unequal variances to test if differences in current exposure are significantly different among areas delivering Low, Medium, and High ecosystem services. This analysis was conducted separately for each of the seven ecosystem services, considering the presence of the subset of species that could potentially impact each service. We added post-hoc Tukey HSD tests to explore pairwise comparisons (Supplementary Table 2). We performed sensitivity analysis using three alternative breakdowns of ecosystem service values that can be consulted in Supplementary Fig. 5.

## Future potential exposure of ecosystem services to invasion

Potential exposure was measured as the total number of invasive species known to affect each ecosystem service that could potentially establish in each 10 × 10 km grid cell in Europe, under a worst-case scenario where they are able to fill the most accessible and favorable parts of their ecological niche. To estimate potential exposure, we used species distribution models (SDMs). These models describe the environmental conditions suitable for the establishment of each invader and provide an estimate of their potential range.

We used 21 variables to model the distribution of each invasive species, including: accessibility (measured by human travel time[52]), elevation[53], and 19 bioclimatic variables obtained from CHELSA (Climatologies at High resolution for the Earth's Land Surface Areas) version 2.1[54]. All variables were downloaded as raster global maps at 0.5–1 arc-minute resolution, and aggregated using the R package 'terra' to 10 arc minutes (~11 × 11 km in central Europe) to approximately match the spatial resolution of the species occurrence data. Pearson's correlations among the 21 variables can be consulted in Supplementary Fig. 7.

Models were built for each invasive species using Bayesian Additive Regression Trees (BART), computed with the 'dbarts' R package v 0.9-21 using the default priors, which have been demonstrated to be remarkably effective[55]. Because BART is robust to the inclusion of redundant variables, we used all predictors in our models. Sensitivity analysis comparing the results of BART with other common algorithms (GLM, GAM, and GBM) and spatial cross-validation can be consulted in Supplementary Data 3. For the assessment of model performance, we used two counter-balancing threshold-independent evaluation metrics: (1) the Area Under the receiver operating characteristic Curve (AUC), which measures the overall discrimination power, i.e., the capacity of the model to assign higher predicted values to presence than to absence localities; (2) Miller's calibration slope (MCS), which measures the reliability of model predictions, i.e., the overall deviation of predicted probabilities from observed occurrence frequencies[56]. An AUC value of 1 indicates perfect discrimination between presences and absences, i.e. that all presences have higher predicted values than any absences. An MCS value of 1 shows that predicted probabilities are directly proportional to observed occurrence frequencies. These metrics were computed with R package 'modEvA' v 3.0. Results from cross-validation indicated good to excellent performance of distribution models, which were fit but not overfit, with an average AUC of $0.86 \pm 0.06$ and MCS of $1.09 \pm 0.18$. More details can be consulted in Supplementary Data 3.

Distribution models provide the probability of establishment of modeled species, which measures how likely we are to find the species at a given place. Probability therefore depends on how good the environment is at that place and how common the species is across the study area. Independently of the environment, presence probability is higher for species that are more common, so species that (still) have few presences, as is normally the case with recently introduced invaders, would be underestimated. For this reason, we used favorability, that is how much the environment favors the presence of each species, after compensating for prevalence[57]. Favorability for each species was classified into three categories: 1 (favorability ≲20%), 2 (20-80% favorability), and 3 (favorability ≳80%). The category 3 represents environments where the odds of favorability for species occurrence are more than 4:1. This classification approach allows us to differentiate areas that strongly favor the establishment of the species from those that are less suitable. It is worth noting that distribution models tend to overestimate the area susceptible for establishment, particularly for species that are restricted to specific habitats (e.g. freshwater), hosts (e.g. forest pests), or for which it is difficult to differentiate casual from established reproducing populations. Hence, we considered that only areas with very high favorability for the species are under the highest threat of invasion, and calculated the potential exposure by aggregating all species with favorability values ≳80%. However, we acknowledge that invasive species may eventually establish in lower favorability areas.

To assess the potential range expansion of the investigated species, we compared the number of 10 × 10 km grid cells in Europe where the species are currently known to occur (current exposure) with the number of cells classified as category 3 in terms of favorability (potential exposure). Differences between Current and Potential were analyzed using a paired t-test (two-sided). Category 3 represents areas that are highly favorable for the establishment of invasive species, and is affected by the threshold used in the classification. Therefore, we conducted the analysis using three alternative thresholds, which are described in detail in Supplementary Methods. By considering different thresholds, we gain insights into the potential range expansion under varying levels of environmental favorability.

### Risks posed by invasive species to ecosystem services

The risk posed by each invasive species on the specific ecosystem services they can affect were estimated by considering: 1) the combination of species favorability for invasion (i.e. exposure), and 2) the delivery of ecosystem services (i.e. vulnerability). Building upon the approach proposed by Pérez et al.[13], we defined a total of nine risk categories by intersecting the provision of ecosystem services (categorized as Low/Medium/High) with the favorability for invasion (categorized 1, 2, and 3; Fig. 4 A). Simply multiplying the favorability for invasive species and the provision of ecosystem services was discarded because it did not allow us to distinguish Safe from Critical areas, leading to a potential misinterpretation of the results.

For the analysis, particular emphasis was put on the following categories:

- *Coldspots* (L1) represent sites where both ecosystem service provisioning and the potential exposure to invasion are low; consequently, impacts associated to invasion are expected to be limited.
- *Critical impact* (L3) represent sites where ecosystem service provisioning is low and the potential exposure to invasion is high; consequently, the invader may compromise entirely the provisioning of the ecosystem services e.g. an invasive species that promotes nutrient accumulation in areas with limited capacity for nitrogen processing may lead to a shift in the ecosystem towards eutrophication, resulting in a complete disruption of ecosystem service provision.
- *Safe provision* (H1) represents sites showing high ecosystem service provisioning and low potential exposure to invasion, where no major problems associated with the invader are expected.
- *Hotspots* (H3) represent sites combining a high delivery of ecosystem services and high potential exposure to invasion. Consequently, this is where we may expect the greatest reduction in the overall provisioning of ecosystem services.

For each of the combinations between 94 invasive species and seven ecosystem services evaluated in this study (total=269), we assessed the extent of Europe classified into the nine risk categories (Supplementary Data 4, and Supplementary Fig. 10). Detailed maps for each combination are available in a Figshare data repository (Figshare, https://doi.org/10.6084/m9.figshare.21739385). To account for the potential influence of different thresholds on the classification of areas into low, medium, and high ecosystem services, we compared three alternative thresholds, which are described in Supplementary Fig. 11.

Finally, we created "density of hotspots" and "density of critical areas" maps by combining hotspot and critical maps for each particular ecosystem service (Fig. 5 and Supplementary Fig. 12). These maps provide insights into the concentration of areas with high ecosystem service provisioning and high potential exposure to invasion; or low ecosystem service provisioning and high potential exposure to invasion, respectively. The values on these maps range from 0 to 100%, representing the degree of overlap between high provisioning areas and areas favorable for invasion. Overall, these analyses contribute to the understanding of the spatial patterns and potential impacts of invasive species on ecosystem service provision in Europe.

### Reporting summary

Further information on research design is available in the Nature Portfolio Reporting Summary linked to this article.

## Data availability

The intermediate data files generated in this study are provided as Datasets in a Figshare repository under accession code https://figshare.com/s/f8b4cf965b71ea5b1577.

## Code availability

The R scripts used in this study are provided in a Figshare repository under accession code https://figshare.com/s/f5bd8287c1e391780879.

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

## Acknowledgements

This research was funded through the 2017–2018 Belmont Forum and BIODIVERSA joint call for research proposals, under the BiodivScen ERANet COFUND program, through the InvasiBES (http://elabs.ebd.csic.es/web/invasibes), with the funding organizations: Spanish State Research Agency (MCI/AEI/FEDER, UE, PCI2018-092939 awarded to M.V. and PCI2018-092986 awarded to B.G.), Swiss National Science Foundation (SNSF grant numbers 31003A_179491 and 31BD30_184114, both awarded to S.B. and G.V.), French National Research Agency (ANR, ANR-18-EBI4-0001-06 awarded to L.G.) and the US National Science Foundation (ICER-1852060 awarded to C.S.). P.G.M. was supported by a

"*Juan de la Cierva-Incorporación*" contract (MINECO,IJCI-2017-31733) and Plan Propio UCO 2020.

## Author contributions

B.G. designed the research, collected and analyzed the data, and led the writing. A.-M.B. and V.M.-B. conducted part of the analyses. S.B., L.G., P.G.-M., C.S., G.V., and M.V. contributed to the conceptual framework and the writing of the study.

## Competing interests

The authors declare no competing interests.

## Inclusion & ethics

The research did not include local researchers or partners that require further description.
