## [Peer Review File · Nature Communications]

Risks posed by invasive species to the provision of ecosystem services in EuropeEditorial Note: This manuscript has been previously reviewed at another journal that is not operating a transparent peer review scheme. This document only contains reviewer comments and rebuttal letters for versions considered at *Nature Communications*.

Reviewer #4 (Remarks to the Author):

This manuscript assesses the potential impact of non-indigenous species on a set of ecosystem services in Europe. The focus is on seven services, determining which services may be affected by the 94 species mapped, and determining where, spatially, there is overlap in the potential for detrimental impacts on services and service hotspots. Results indicate limited impact on most high-service producing areas. Mapping extent is Western Europe and resolution is 10km x 10km. An earlier version of this manuscript had already undergone review, with revision and responses. I do not repeat comments of previous reviews unless the response needed further clarification or comment.

First, the overall project is of interest, impacts of non-indigenous species currently or potentially established is important, given the largely detrimental impact of nonindigenous species. Although impacts on ES have been previously investigated the extent and scope makes this manuscript interesting. Having said that, my focus below is on ways to improve the manuscript as currently presented.

First, the term invasive alien species is poor and should not be used. In the USA, a presidential executive order defining invasive species as "exotic" and economically costly is an administrative definition, not a biological one. and the terms exotic and alien are both "loaded". The paper deals with non-indigenous or non-native species, that may have been/be introduced or otherwise invade, and may or may not be "invasive" in a particular setting. So it would be better to use more biologically relevant terms, such as non-native or non-indigenous species.

I believe the subjective categorization of species and their impacts includes much uncertainty, and that uncertainty can propagate in the authors' calculations. There is uncertainty in the bin membership, in the species models, in the ES production, among other sources. This should be fully acknowledged and addressed. The potential for non-linear response as raised earlier needs to be better addressed.

Given the great uncertainty discussed above, combined with the very coarse resolution, I think the assessment is "liberal" rather than conservative. I think that is OK, when searching for broad patterns it is better to at first reduce Type II errors rather than Type I errors, and refine assessments further if those patterns are identified. But I think conservative is wrong given the presentation of the results. That is, I think it overestimates impacts.

Given the high uncertainty in many elements of the study, why not test some of the results in a limited number of 10x10 grid cells with better models of higher resolution for both non-indigenous species and for ES? I suspect the error rate would be extremely high, as the authors acknowledge with aquatic species.

Often, rather than simply a loss of ES, we are looking at tradeoffs in the suites of services with change, and those tradeoffs vary with scale. It would be very good to see scale addressed here, and there are many aspects - the geographic scale of the models, the scale specific impacts, the potential for impacts to cascade up scales, and more.

The authors discuss the fact that exposure is highest in areas with low ES, but don't really address why this may be. I assume it is because ES production and non-indigenous species occurrence are both highest in disturbed habitats or more anthropogenic habitats. Discussion would be nice.

Reviewer #5 (Remarks to the Author):

Dear Authors,

I first read the revised manuscript, and then the rebuttal letter. I agree with the two first reviewers that this is an interesting and innovative paper. I am not an expert on ES and NCP, though I think the authors convincingly justify their approach to scoring the potential impact on invasive species. Data on abundance and per capita impacts of invasive species on ES/NCP are just not available at larger spatial scales, and the categorization used by the authors is I think a valid way of exploring what the data and analyses can learn us about current and future IAS impacts.

Lines 384-385: Author reply to Question 6 of Reviewer #1. I would recommend adding a succinct version of this justification for using global models directly in the ms. I think it is important to clarify directly in the manuscript that both native and invasive range occurrence data are used in order to as fully as possible 'map' the species' environmental niche. This has relevance to the questions of reviewer #3 about data reliability, with special reference to the invasive range data (see below).
Lines 394-400: thus background area = rarefied occurrence datapoints buffered with 200km? From the quoted Gallien et al. 2012 paper: "In order to restrict the choice of pseudo-absences to realistically reachable locations, we created a buffer zone of 20 km around any of the presence records used and we randomly allocated absences inside these buffer zones. In this way, we avoid areas where invasive species have not been inventoried." I think that the current wording of this part could be improved and that the Liang et al. (2018) paper mentioned in the reply to Question 3 of Reviewer#2 should be cited here.

Line 492: cubes?

Sup Fig 2 illustrates the issues with data reliability, cfr. the 'maps like the ones produced in this paper are only as good as their underlying data' in Question #6 of Reviewer #3. Using global occurrence data is indeed a tested procedure for forecasting invasion risks, but occurrence data such as those used here have many potential biases and data quality issues. The authors mostly carried out common data cleaning procedures to catch the worst errors and their pseudo-absence selection strategy is aimed at reducing the influence of sampling bias. The SDM methods are sound and clearly explained. However, Sup Fig 2 shows the European occurrence data used to train the models, revealing that for species such as the birds *Acridotheres cristatellus*, *Acridotheres tristis*, and *Corvus splendens*, but also for the chipmunk *Tamias sibiricus*, there are plenty of occurrence data that do not reflect self-sustaining native populations. This may also be the case for some other species modeled, but I am only familiar with the EU distribution of birds and mammals.

Deciding which occurrences reflect self-sustaining populations is fraught (e.g. by comparing GBIF occurrence data with recent published breeding bird atlases etc.). Indeed, for many taxa, such comparisons are not even possible because of a lack of data except for the observation of the species at a given place and time). Invasive range data are included sensu Broennimann & Guisan (2008) to better approximate the species' environmental tolerances, but including occurrence data that do not reflect established populations risks overestimating species' niches. These occurrences typically reflect animals still commonly held in captivity, and only surviving for a certain period without (successful) reproduction.

Thus, a presence used to train the models can either indicate 1) the environment allows for the persistence of the species, or 2) the species is 'present' without (successful) reproduction but only because new individuals are regularly released by humans. Is it not possible that because of accepting 'all' invasive range data for model training, the SDMs overestimate invasion risk?

Fig S10: This is an example where, maybe because of formatting issues, it is a bit difficult to read the labels on figures because of small font and low resolution.

RESPONSE TO REVIEWERS

This file contains the response to reviewers corresponding to the following manuscript:

Risks posed by invasive species to the provision of ecosystem services

By Belinda Gallardo and colleagues (belinda@ipe.csic.es)

NOTE: Line numbers refer to the clean manuscript (without track changes).

Reviewer #4 (Remarks to the Author):

This manuscript assesses the potential impact of non-indigenous species on a set of ecosystem services in Europe. The focus is on seven services, determining which services may be affected by the 94 species mapped, and determining where, spatially, there is overlap in the potential for detrimental impacts on services and service hotspots. **Results indicate limited impact on most high-service producing areas.** Mapping extent is Western Europe and resolution is 10km x 10km. An earlier version of this manuscript had already undergone review, with revision and responses. I do not repeat comments of previous reviews unless the response needed further clarification or comment.

First, the overall project is of interest, impacts of non-indigenous species currently or potentially established is important, given the largely detrimental impact of nonindigenous species. Although impacts on ES have been previously investigated the extent and scope makes this manuscript interesting. Having said that, my focus below is on ways to improve the manuscript as currently presented.

Response: Thank you for your thoughtful and positive comments regarding the significance of our manuscript. We appreciate your recognition of the importance of studying the impacts of invasive species on ecosystem services, and we value your acknowledgment of the extensive scope and interest of our research project.

First, the **term invasive alien** species is poor and should not be used. In the USA, a presidential executive order defining invasive species as "exotic" and economically costly is an administrative definition, not a biological one. and the terms exotic and alien are both "loaded". The paper deals with non-indigenous or non-native species, that may have been/be introduced or otherwise invade, and may or may not be "invasive" in a particular setting. So it would be better to use more biologically relevant terms, such as non-native or non-indigenous species.

Response. Regarding the terminology concern, we acknowledge your perspective on the term "invasive alien species" and agree with the need for clarity, particularly for US readers. However, we should point out that "Invasive Alien Species" (IAS) is the term: i) adopted by the Intergovernmental Science-Policy Platform on Biodiversity and Ecosystem Services (IPBES) and used in its Global Assessment Report (IPBES, 2023), accepted by 143 member states (including the US), ii) used by the IUCN and the EICAT Authority that revises all global risk assessments for invasive species, and ii) officially employed by the European Commission in its IAS Regulation 1143/2014, and the Union List of IAS of Concern, that we used as focus list.

In addition, we'd like to point out that the focus species are not just "non-native", but that, based on the scientific evidence available, all of them have been classified as invasive, a prerequisite to enter the Union List. So, after careful consideration and consultation among co-authors, we agreed to use "invasive species" throughout the manuscript, including the title, thereby dropping the "alien" part that is possibly the most problematic. We believe this approach is a balance that respects the reviewer's input while maintaining clarity and coherence of our research with international organizations.

I believe the subjective categorization of species and their impacts includes much uncertainty, and that **uncertainty can propagate in the authors' calculations**. There is uncertainty in the bin membership, in the species models, in the ES production, among other sources. This should be fully acknowledged and addressed. **The potential for non-linear response as raised earlier needs to be better addressed.**

Response: We appreciate your thoughtful comments regarding the uncertainty inherent in our study. We fully acknowledge that the categorization of species and their impacts introduces an element of subjectivity, and this uncertainty can propagate throughout our calculations. However, it's important to note that the challenges associated with uncertainty in macroecological studies are not unique to our work. Many studies in this field have to make similar simplifications when dealing with large-scale ecological processes and species interactions. We have tried to address these uncertainties to the best of our ability within the constraints of the word limitations of the journal.

Lines 218-220: "Sensitivity analysis using alternative thresholds for service provisioning showed variation in the classification of Hotspots (0.6% to 1.8% of Europe) and Critical areas (0.7% to 2.3% of Europe), while Safe areas ranged from 6.4% to 17.7% of Europe (Fig. S11)."

Lines 308-328: "However, it is important to acknowledge that our assessment has limitations in fully capturing the magnitude of impacts and their context dependence. The mechanisms driving the impacts of invasive species on ecosystem services are complex, and empirical evidence is often limited. The presence of an invasive species does not necessarily guarantee an impact on ecosystem services, as factors such as invader density and management measures can influence the outcomes. Uncertainty in our research arises from several factors, including the association between invasive species and ecosystem services, the modeling of species, the production of ecosystem services, and the thresholds used to define high exposure and vulnerability levels. We have incorporated sensitivity analyses to address and quantify this uncertainty, which have not indicated significant variability in our results. As the availability of quantitative information regarding the relationship between the abundance of invasive species and their impacts of ecosystem functioning and services continues to grow, future research should build upon our initial results. This research should explore how changes in one ecosystem service, due to the presence of invasive species, have cascading effects on other services across scales (e.g. Walsh et al. 2016). Additionally, it is important to consider the cumulative or synergistic consequences of multiple invasive species and how they collectively impact ecosystem services. Indeed, our analysis made no assumptions regarding the additive or synergistic impacts of co-occurring invasive species. However, interactions deserve further investigation, as the combined presence of multiple invasive species could exacerbate their individual impacts, potentially leading to a threshold where ecosystem functioning becomes severely compromised. By considering these connections, we can gain a more comprehensive understanding of the broader socio-ecological consequences of biological invasions."

Lines 369-373: “Consequently, we adopted a precautionary approach and categorized potential impacts as either “yes” or “no”. It is important to note that: 1) the magnitude of impacts can vary depending on the specific context, making it difficult to extrapolate values to the entire affected area, 2) assessors' information sometimes lacked the necessary details to estimate individual impact magnitudes, and 3) certain invaders are not yet present in Europe, leading to uncertainty regarding their potential impacts.”

Lines 381-385: “However, it must be noted that GBIF records do not differentiate casual from established self-sustaining populations, so that they may overestimate the current spread of invaders. However, following a precautionary principle, it is preferable to overestimate rather than underestimate the current spread of invasive species, since casual populations may still exert the same impacts on biodiversity and ecosystem services.”

Lines 417-419: “Please note that the reference year for the mapping of ecosystem services is 2012, which means that the level of provision incorporates the historical influence of various threats, including biological invasions.”

Lines 428-436: “Using categories of ecosystem service provision instead of raw values is justified for several reasons. First, the classification allows for a more intuitive interpretation of the data, as it simplifies complex and potentially variable values into distinct categories. Second, the absolute thresholds for high or low provisioning are not well-defined or universally agreed upon. By using percentiles, such as the upper and lower 20%, we can create categories that capture the relative differences in provisioning levels within the dataset. Third, the use of categories acknowledges that ecosystem service provision is not equally distributed across the study area. Categorization allows us to identify areas of particularly high or low provisioning relative to the overall distribution, providing valuable insights for management and decision-making.”

Lines 439-447: “Current exposure was quantified as the total number of invasive species affecting a specific ecosystem service within each grid cell, without considering their abundance, which may vary considerably seasonally across their distribution range. Several factors justify our approach. First, the relationship between abundance and impact is non-linear, and even low abundances can disrupt ecosystem functions (Bradley et al. 2019). Second, information about abundance data is rarely available at large spatial scales like the one used in this study. Third, abundance of invasive species generally increases over time (Simberloff et al. 2013), which means that using current or historical abundance would underestimate the potential impacts of invaders. Considering these complexities, we believe that our approach is appropriate for an initial approximation.”

Lines 497-500: “It is worth noting that distribution models tend to overestimate the area susceptible for establishment, particularly for species that are restricted to specific habitats (e.g. freshwater) or hosts (e.g. forest pests), or for which it is difficult to differentiate casual from established reproducing populations.”

Given the great uncertainty discussed above, combined with the very coarse resolution, I think the assessment is "liberal" rather than conservative. I think that is OK, when searching for broad patterns it is better to at first reduce Type II errors rather than Type I errors, and refine assessments further if those patterns are identified. But I think conservative is wrong given the presentation of the results. That is, I think it **overestimates impacts**.

Response: Your point about prioritizing the reduction of Type II errors in the initial phase of searching for broad patterns aligns with our overarching goal of identifying and understanding potential impacts. To address your concern about the term "conservative," we have carefully reconsidered its usage in our manuscript, and deleted it when necessary. It is crucial that our presentation accurately reflects the nature of our assessment and the associated uncertainties.

Given the high uncertainty in many elements of the study, why not **test some of the results** in a limited number of 10x10 grid cells with better models of higher resolution for both non-indigenous species and for ES? I suspect the error rate would be extremely high, as the authors acknowledge with aquatic species.

Response: We appreciate the thoughtful suggestion to test our results in a limited number of 10x10 grid cells with higher-resolution models for invasive species and ecosystem services (ES). While we acknowledge the potential benefits of such an approach, we respectfully provide two main reasons for not following this suggestion. First, accurate high-resolution data for both invasive species and ecosystem services are currently lacking, especially at a scale that aligns with the purpose of the study. Second, the scope of our analysis is designed to provide a comprehensive overview of the potential impacts of invasive species on ecosystem services across Europe. Testing specific grid cells with higher-resolution models would deviate from the broader regional focus and introduce additional complexities that extend beyond the objectives of our study. We have, however, explicitly mentioned the need to test the results from our research at other scales and contexts (see Lines 308-328 referenced above).

Often, rather than simply a loss of ES, we are looking at tradeoffs in the suites of services with change, and those tradeoffs vary with scale. It would be very good to see scale addressed here, and there are many aspects - the geographic scale of the models, the scale specific impacts, the potential for impacts to cascade up scales, and more.

Response: In the revised version, we have addressed the potential for these impacts to cascade across different scales.

See Lines 308-328 referenced above.

The authors discuss the fact that exposure is highest in areas with low ES, but don't really address why this may be. I assume it is because ES production and non-indigenous species occurrence are **both highest in disturbed habitats or more anthropogenic habitats**. Discussion would be nice.

Response: We appreciate the reviewer's observation regarding the correlation between high exposure and low ecosystem service production in certain areas. This pattern is indeed a focal point of our study, and we acknowledge its significance. The manuscript explores how the occurrence of invasive species is often concentrated in human-altered habitats, leading to a potential decrease in ecosystem service production. In the revised manuscript, we have introduced this question in the introduction and amplified it in the discussion. We thank the reviewer for drawing attention to this aspect.

Lines 77-84: "For instance, human-altered habitats are highly exposed to invasions because of their high levels of disturbance and propagule pressure (Gallardo et al 2015, 2017). This is frequently the case for invasive plants, whereas invasive animals tend to be more evenly

distributed in natural and human-altered environments (Liu et al. 2023, Pysek et al. 2010). However, human-altered habitats are typically unable to deliver a wide variety of ecosystem services (IPBES 2019), therefore displaying limited vulnerability. In contrast, well-preserved areas largely contributing to many different services may be less exposed to the current and potential threats of biological invasions, but are highly vulnerable, making any future changes significantly impactful (Gallardo et al. 2017)”.

Lines 293-296: “Indeed, it is well-established that human-altered habitats are more susceptible to invasions due to their elevated levels of disturbance and propagule pressure (Gallardo et al. 2015, but see Pysek et al. 2010). Such habitats are also known to provide fewer ecosystem services, or more concentrated in provisioning services, such as energy, food and timber (IPBES 2019).”

Lines 452-454: “We expected significantly higher pressure associated with biological invasions in low-ecosystem service delivering areas, typically characterized by altered habitats with high propagule pressure and invasibility.”

Reviewer #5 (Remarks to the Author):

Dear Authors,

I first read the revised manuscript, and then the rebuttal letter. I agree with the two first reviewers that this is an interesting and innovative paper. I am not an expert on ES and NCP, though I think the authors convincingly justify their approach to scoring the potential impact on invasive species. Data on abundance and per capita impacts of invasive species on ES/NCP are just not available at larger spatial scales, and the categorization used by the authors is I think a valid way of exploring what the data and analyses can learn us about current and future IAS impacts.

Response: Thank you for acknowledging the innovative approach and justifications presented in the manuscript. We appreciate your understanding of the challenges associated with obtaining data on abundance and per capita impacts of invasive species on ecosystem services at larger spatial scales.

Lines 384-385: Author reply to Question 6 of Reviewer #1. I would recommend adding a succinct version of this justification for using global models directly in the ms. I think it is important to clarify directly in the manuscript that both native and invasive range occurrence data are used in order to as fully as possible ‘map’ the species' environmental niche. This has relevance to the questions of reviewer #3 about data reliability, with special reference to the invasive range data (see below).

Response: We have included this in the new version:

Lines 379-385: “Relying on global occurrence data is essential, ensuring coverage of the full environmental niche and avoiding underestimation, especially for species not yet in Europe or recently established (Broennimann & Guisan 2008; Jiménez-Valverde et al. 2011). However, it must be noted that GBIF records do not differentiate casual from established self-sustaining populations, so that they may overestimate the current spread of invaders. However, following a precautionary principle, it is preferable to overestimate rather than underestimate or bias the

spread of invasive species, since casual populations may still exert the same impacts on biodiversity and ecosystem services.”.

Lines 394-400: thus background area = rarefied occurrence datapoints buffered with 200km? From the quoted Gallien et al. 2012 paper: “In order to restrict the choice of pseudo-absences to realistically reachable locations, we created a buffer zone of 20 km around any of the presence records used and we randomly allocated absences inside these buffer zones. In this way, we avoid areas where invasive species have not been inventoried.” I think that the current wording of this part could be improved and that the Liang et al. (2018) paper mentioned in the reply to Question 3 of Reviewer#2 should be cited here.

Response: We have clarified this question in the text.

Lines 399-403: “Pseudo-absences were generated by randomly selecting separate 10-arc-minute pixels, in the same number of each species’ presences, outside the presence pixels but within a 200-km spatial buffer around them. This aims to restrict the choice of pseudo-absences to potentially reachable areas in regions where the species has been surveyed, thus conferring pseudo-absences the same survey bias as presences, which is key to improve the transferability of models (Liang et al. 2018; Gallien et al. 2012).”

Sup Fig 2 illustrates the issues with data reliability, cfr. the ‘maps like the ones produced in this paper are only as good as their underlying data’ in Question #6 of Reviewer #3. Using global occurrence data is indeed a tested procedure for forecasting invasion risks, but occurrence data such as those used here have many potential biases and data quality issues. The authors mostly carried out common data cleaning procedures to catch the worst errors and their pseudo-absence selection strategy is aimed at reducing the influence of sampling bias. The SDM methods are sound and clearly explained. However, Sup Fig 2 shows the European occurrence data used to train the models, revealing that for species such as the birds *Acridotheres cristatellus*, *Acridotheres tristis*, and *Corvus splendens*, but also for the chipmunk *Tamias sibiricus*, there are plenty of **occurrence data that do not reflect self-sustaining native populations**. This may also be the case for some other species modeled, but I am only familiar with the EU distribution of birds and mammals. Deciding which occurrences reflect self-sustaining populations is fraught (e.g. by comparing GBIF occurrence data with recent published breeding bird atlases etc.). Indeed, for many taxa, such comparisons are not even possible because of a lack of data except for the observation of the species at a given place and time). Invasive range data are included sensu Broennimann & Guisan (2008) to better approximate the species’ environmental tolerances, but including **occurrence data that do not reflect established populations** risks overestimating species’ niches. These occurrences typically reflect animals **still commonly held in captivity**, and only surviving for a certain period without (successful) reproduction. Thus, a presence used to train the models can either indicate 1) the environment allows for the persistence of the species, or 2) the species is ‘present’ without (successful) reproduction but only because new individuals are regularly released by humans. Is it not possible that because of accepting ‘all’ invasive range data for model training, the SDMs overestimate invasion risk?

Response: Indeed, at this spatial scale, for most species (even those with distribution atlas) it is impossible to determine which occurrences reflect self-sustaining populations. In these circumstances, we believe it is better to be conservative and keep all reliable presences (even if some represent non-reproducing populations) than to risk underestimating, or worse, bias the invasive potential. We now mention this in the manuscript:

Please see Lines 379-385 referenced above.

Fig S10: This is an example where, maybe because of formatting issues, it is a bit difficult to read the labels on figures because of small font and low resolution.

Response: We have increased the font size to facilitate interpretation and have provided databases and R scripts as supplementary material, allowing replication of the graph.

Reviewer #4 (Remarks to the Author):

These comments are on the revised version of the ms "Risks posed by invasive species to the provision of ecosystem services". I won't provide any additional summary (in earlier review) but focus on responses to that earlier review, and a read of the manuscript.

- I'm happy with the elimination/reduction of the term "alien", not appropriate despite use of the term by governments.

- Uncertainty. I'm a bit confused by the sensitivity analysis here - usually used to determine variables most affecting the outcome, but here seems to be simply assessing variation in results. I think provided variability information is good, and important, but not really a sensitivity analysis. Authors state that magnitude of impacts is problematic, but it's even more than that - - there's uncertainty regarding whether an impact is even present - co-occupying 10x10km grids provides the potential for impact, but at such large spatial scales, the non-native sp. and the ES of interest may not interact. And there's still a lot of other sources of uncertainty - the models themselves, the data, assumptions of co-occupancy at relevant scales, etc. Although uncertainty is always present, there have been solid efforts to quantify uncertainty in similar modelling processes.

Regardless of that, I still think being liberal is fine - worst case scenario that all the interactions assumed to be happening actually are. And for abundance, it's not just that pops can vary seasonally, but abundance is really unknown, and there can be thresholds in abundance before there are deleterious impacts.

- regarding disturbed habitats. Seems like authors mental model is altered habitat - invasions - impact on ES. Could also be the altered habitats have simultaneous impacts on invasions and ES and that these three elements further interact to degrade the systems in question.

Some other comments from reading ms:

- large scales - - this is really large spatial scales.

- line 54 - specific services = 7 services

- line 61 "a" continental scale

- 62 - synthesize - really, map.

- 68 - seems the worst case scenario would be invasive species occupying unfavorable parts of their niche. There is also the assumption that these spp and ES interact meaningfully at 10x10 km scales.

76 - "well-preserved" - conserved? Managed?

77 - why are these "highly vulnerable"?

123 - presence does not equal exposure, at these spatial scales.

133 - "growing request"?

311, 312 - results show a 3x difference given variability, seems pretty significant.

(Remarks on code availability):

Not really able to comment.

Reviewer #5 (Remarks to the Author):

I have checked the responses of the authors to my comments, both in the reply letter and the manuscript itself, in detail. The authors have responded to my questions in a satisfactory manner and I have no further questions. It is good that the paper now explicitly mentions sources of uncertainty that can either lead to over- or underestimating risk. The authors now make clear that where uncertainty exists, they apply a precautionary approach and that allows readers to judge the results

for themselves in a transparent manner.

The other reviewer has very pertinent remarks about source of uncertainty and whether they lead to over- or underestimation, but the authors are also right that these issues are not unique to this manuscript. Indeed, most modeling studies are faced with this, and I am of the opinion that the current ms does a good job (and better than most papers) in trying to account for them, and transparently mentioning them.

I hence have no objections to seeing this manuscript published.

Best regards,

Diederik Strubbe

RESPONSE TO REVIEWERS

This file contains the response to reviewers corresponding to the following manuscript:

Risks posed by invasive species to the provision of ecosystem services in Europe

By Belinda Gallardo and colleagues (belinda@ipe.csic.es)

Reviewer #4 (Remarks to the Author):

These comments are on the revised version of the ms "Risks posed by invasive species to the provision of ecosystem services". I won't provide any additional summary (in earlier review) but focus on responses to that earlier review, and a read of the manuscript.

- I'm happy with the elimination/reduction of the term "alien", not appropriate despite use of the term by governments.

- Uncertainty. I'm a bit confused by the sensitivity analysis here - usually used to determine variables most affecting the outcome, but here seems to be simply assessing variation in results. I think provided variability information is good, and important, but not really a sensitivity analysis. Authors state that magnitude of impacts is problematic, but it's even more than that - - there's uncertainty regarding whether an impact is even present - co-occupying 10x10km grids provides the potential for impact, but at such large spatial scales, the non-native sp. and the ES of interest may not interact. And there's still a lot of other sources of uncertainty - the models themselves, the data, assumptions of co-occupancy at relevant scales, etc. Although uncertainty is always present, there have been solid efforts to quantify uncertainty in similar modelling processes.

Response: Thank you for your thoughtful comments regarding our sensitivity analysis and the associated uncertainties in our study. We agree that our study, like all modeling studies, is subject to various sources of uncertainty, including (but not limited to) biases in occurrence data, quantification of ecosystem services, and assumptions regarding co-occupancy at relevant scales. We have made substantial efforts to acknowledge these limitations to the best of our ability, underscoring the inherent challenges in modeling the context-dependent impacts of invasive species.

We understand the confusion around our use of the term 'sensitivity analysis.' Our intent was to assess the variability in outcomes as a direct response to changes in specific model parameters that we arbitrarily set, notably the threshold used to categorize continuous variables. We also investigated the importance of variables to determine the species occurrent, which is reported in Supplementary Figure 8.

We recognize your point regarding the potential lack of interaction between non-native species and ES of interest within large 10x10km cells. This is specifically mentioned in the manuscript.

Lines 307-314: *“The presence of an invasive species does not necessarily guarantee an impact on ecosystem services, as factors such as invader density, microhabitat and management measures can influence the outcomes. Uncertainty in our research arises from several factors, including the association between invasive species and ecosystem services, map resolution, the modeling of species, the production of ecosystem services, and the thresholds used to define high exposure and vulnerability levels. We have incorporated sensitivity analyses to address and quantify part of this uncertainty, which have not indicated significant variability in our results.”*

Regardless of that, I still think being liberal is fine - worst case scenario that all the interactions assumed to be happening actually are. And for abundance, it's not just that pops can vary seasonally, but abundance is really unknown, and there can be thresholds in abundance before there are deleterious impacts.

Response: Agreed, we have incorporated this in the text:

Lines 439-441 : *“ the relationship between abundance and impact is non-linear; in some cases, there can be thresholds in abundance before there are deleterious impacts, whereas in others even low abundances can disrupt ecosystem functions”.*

- regarding disturbed habitats. Seems like authors mental model is altered habitat - invasions - impact on ES. Could also be the altered habitats have simultaneous impacts on invasions and ES and that these three elements further interact to degrade the systems in question.

Response: Agree, while it is reasonable to expect a high level of propagule pressure associated to altered habitats, and therefore impact on ES, this is evidently an oversimplification. Our results suggest that this may be true for some species and services (e.g. crop provision, soil retention) but not all. The fact that the ES maps already reflects the effect of environmental degradation is acknowledged in;

Lines 415-417: *“Please note that the reference year for the mapping of ecosystem services is 2012, which means that the level of provision incorporates the historical influence of various threats, including biological invasions.”*

All minor comments have been tackled in the text:

Some other comments from reading ms:

- large scales - - this is really large spatial scales.
- line 54 - specific services = 7 services
- line 61 "a" continental scale
- 62 - synthesize - really, map.
- 68 - seems the worst case scenario would be invasive species occupying unfavorable parts of their niche. There is also the assumption that these spp and ES interact meaningfully at 10x10 km scales.
- 76 - "well-preserved" - conserved? Managed?
- 77 - why are these "highly vulnerable"?
- 123 - presence does not equal exposure, at these spatial scales.
- 133 - "growing request"?
- 311, 312 - results show a 3x difference given variability, seems pretty significant.

Reviewer #5 (Remarks to the Author):

I have checked the responses of the authors to my comments, both in the reply letter and the manuscript itself, in detail. The authors have responded to my questions in a satisfactory manner and I have no further questions. It is good that the paper now explicitly mentions sources of uncertainty that can either lead to over- or underestimating risk. The authors now make clear that where uncertainty exists, they apply a precautionary approach and that allows readers to judge the results for themselves in a transparent manner.

The other reviewer has very pertinent remarks about source of uncertainty and whether they lead to over- or underestimation, but the authors are also right that these issues are not unique to this manuscript. Indeed, most modeling studies are faced with this, and I am of the opinion that the current ms does a good job (and better than most papers) in trying to account for them, and transparently mentioning them.

I hence have no objections to seeing this manuscript published.

Best regards,

Diederik Strubbe

Response: We sincerely appreciate your thorough examination of our responses and the manuscript revisions. Your constructive feedback has been instrumental in enhancing the quality and clarity of our paper.

Thank you for your time and valuable contributions to this research.